# Maternal anaemia during early pregnancy and the risk of neonatal outcomes: a prospective cohort study in Central China

Yige Chen,[1,2] Taowei Zhong,[1] Xinli Song,[1] Senmao Zhang,[1] Mengting Sun,[1] Xiaoying Liu,[3,4] Jianhui Wei,[1] Jing Shu,[1] Yiping Liu,[1] Jiabi Qin [1,5]

YC and TZ contributed equally.

For numbered affiliations see end of article.

**Correspondence to**
Yiping Liu; 1064960669@qq.com

Professor Jiabi Qin; qinjiabi123@163.com

## ABSTRACT

**Background** The purpose of this study was to explore the association between anaemia during early pregnancy and the risk of neonatal outcomes.

**Methods** We collected clinical data from pregnant women (≥18 years) who received their first antenatal care between 8 and 14 weeks of gestation in Hunan Provincial Maternal and Child Health Care Hospital. Multiple logistic regression models and restricted cubic spline regression models were used to analyse the association between anaemia during early pregnancy and the risk of neonatal outcomes. In addition, sensitivity analysis was further performed to assess the robustness of the results.

**Results** The prospective cohort study ultimately included 34 087 singleton pregnancies. In this study, the rate of anaemia during early pregnancy was 16.3%. Our data showed that there was a positive relationship between the rate of preterm birth, low birth weight as well as small for gestational age (SGA) and the severity of maternal anaemia ($P_{trend}$<0.05). After adjustment, the association of early pregnancy anaemia and haemoglobin (Hb) levels with the risk of preterm birth (mild anaemia adjusted OR (aOR) 1.37 (95% CI 1.25 to 1.52), moderate anaemia aOR 1.54 (95% CI 1.35 to 1.76) and severe anaemia aOR 4.03 (95% CI 2.67 to 6.08), respectively), low birth weight (mild anaemia aOR 1.61 (95% CI 1.44 to 1.79), moderate anaemia aOR 2.01 (95% CI 1.75 to 2.30) and severe anaemia aOR 6.11 (95% CI 3.99 to 9.36), respectively) and SGA (mild anaemia aOR 1.37 (95% CI 1.25 to 1.52), moderate anaemia aOR 1.54 (95% CI 1.35 to 1.76) and severe anaemia aOR 2.61 (95% CI 1.74 to 4.50), respectively; $P_{non-linear}$<0.05) was observed. However, no association was found between early pregnancy anaemia or Hb levels and the risk of congenital malformations. Sensitivity analysis verified the stability of the results.

**Conclusions** Maternal anaemia during early pregnancy was associated with an increased risk of preterm birth, low birth weight and SGA and their rates may increase with the severity of maternal anaemia.

**Trial registration number** ChiCTR1800016635.

## WHAT IS ALREADY KNOWN ON THIS TOPIC

⇒ Anaemia is the most common nutritional deficiency in pregnancy across the world.
⇒ Maternal anaemia is related to adverse neonatal outcomes, but there are inconsistent findings.
⇒ These associations of maternal anaemia with adverse neonatal outcomes may vary when the severity of gestational anaemia is considered.

## WHAT THIS STUDY ADDS

⇒ Maternal anaemia in the first trimester (8–14 weeks) is associated with an increased risk of preterm birth, low birth weight and small for gestational age (SGA).
⇒ There is a positive relationship between the rate of preterm birth, low birth weight as well as SGA and the severity of maternal anaemia.
⇒ Non-linear associations of maternal haemoglobin concentrations with risk of preterm birth, low birth weight and SGA in offspring are observed.

## HOW THIS STUDY MIGHT AFFECT RESEARCH, PRACTICE OR POLICY

⇒ This study may ensure the reliability of the relationship between maternal anaemia and adverse neonatal outcomes.
⇒ To involve pregnant women with different severity of anaemia in timely prenatal care may be beneficial to reducing the risk of adverse neonatal outcomes.

than established threshold values, thereby impairing the capacity of the oxygen-carrying to tissues.[1] It has been claimed to be the most common nutritional deficiency in pregnancy across the world, bringing a high burden to developed and developing countries.[2,3] Anaemia in pregnancy constitutes a global public health challenge with its high prevalence. Stevens *et al*[4] reported that the global prevalence of anaemia in pregnant women was 38%, affecting approximately 32 million individuals. The significance of reducing and properly managing maternal anaemia has been recognised globally.

## INTRODUCTION

Anaemia is a pathophysiological condition in which the level of haemoglobin (Hb) and/or the quantity of red blood cells are lower

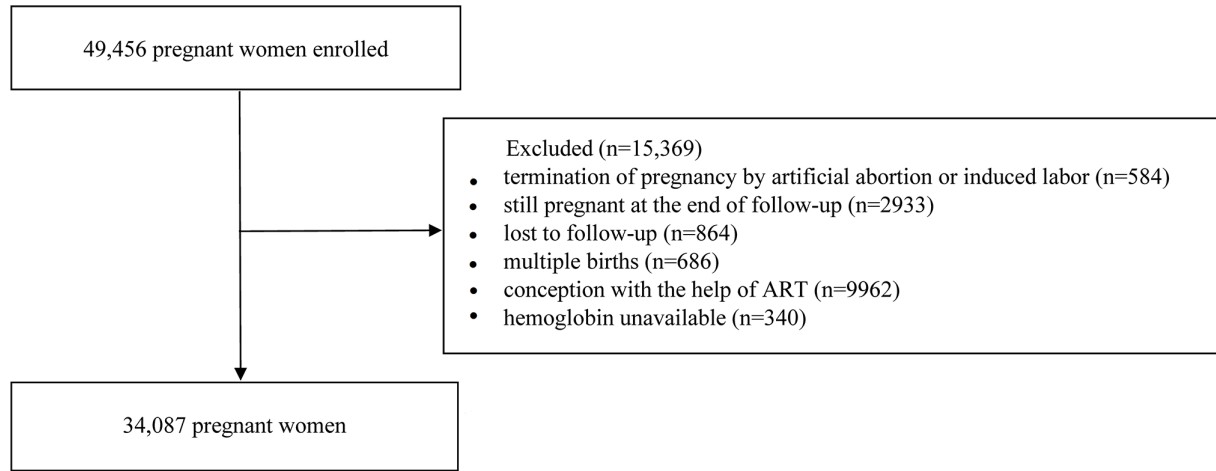

**Figure 1** Flow chart showing the process of participant's recruitment. ART, assisted reproductive technology.

Current targets which are set in Sustainable Development Goal 3 and[5] Global Nutrition Targets 2025 call for a 50% decrease of anaemia in women of reproductive age by the year 2025.[5]

Anaemia during pregnancy is a key contributor to adverse neonatal outcomes.[6] Findings have widely demonstrated associations between maternal anaemia and adverse neonatal outcomes, such as low birth weight, preterm birth and small for gestational age (SGA).[7 8] But these associations may be different when the severity of gestational anaemia is taken into account. Limited data suggested that risks of low birth weight decreased in the offspring of pregnant women who have mild anaemia compared with pregnant women who have no anaemia.[9] A study of 421 pregnant women in India indicated that an association between the severity of anaemia and low birth weight as well as SGA.[10]

The Hb concentration of pregnant women decreases from the second trimester.[11] This is due to the reason that the expansion of plasma volume begins in the second trimester of pregnancy physiologically.[12] Therefore, measurements of Hb made during or after the second trimester are likely to overvalue the prevalence of anaemia. The level of Hb estimated before the 14th week of gestation may reflect Hb levels in pre-pregnancy women and guide future timing and delivery of maternal interventions and prenatal care.[13] In addition, organogenesis occurs in first-trimester pregnancy, during which embryos and fetuses are liable to intense epigenetic reprogramming. It has been suggested that gestational complications, intrauterine environments and a maternal inadequate nutritional status could ultimately result in a range of adverse neonatal outcomes via epigenetic change.[14–16] What is more, anaemia during the first trimester was more detrimental.[17] Overall, anaemia in the first trimester of pregnancy is more likely to lead to adverse neonatal outcomes and is more harmful than anaemia in other trimesters. So, we took maternal anaemia in the first trimester for analysis in this study.

To sum up, this study was based on a prospective cohort study to further clarify the association between the severity of anaemia during early pregnancy and the risk of adverse neonatal outcomes in China.

## METHODS
### Recruitment of study participants
A prospective cohort study design was implemented. From 13 March 2013 to 31 January 2020, pregnant women (≥18 years) who received their first antenatal care at 8–14 weeks of gestation with the intention of continuously receiving prenatal care throughout gestation at Hunan Provincial Maternal and Child Health Care Hospital, in Hunan Province, China were approached and recruited into this cohort. Gestational weeks were based on the last menstrual period data, or estimated by ultrasonography if menstruation was irregular. During these pregnant women's first antenatal visit, a total of 49 456 pregnant women who met these inclusion criteria were invited to be involved in this present cohort. Considering assisted reproductive technology (ART) may lead to misestimation of the prevalence of preterm birth and low birth weight, women concepted with the help of ART were not included in our study. We excluded 15 369 women on the strength of the following exclusion criteria: termination of pregnancy by artificial abortion or induced labour (n=584); still pregnant at the end of follow-up (n=2933); multiple births (n=686); conception with the help of ART (n=9962); lost to follow-up (n=864); Hb unavailable (n=340). Finally, a total of 34 087 women were included in the current analysis. Detailed recruitment criteria of study population were shown in figure 1.

Additionally, it had been registered in the Chinese Clinical Trial Registry Center (date of registration: 14 June 2018). Information and biological samples were collected from participants after obtaining written informed consent from all participants.

### Information collection
After recruitment, the specially trained investigators used a structured questionnaire to collect information on maternal socio-demographic characteristics through

**Table 1** Distribution of maternal characteristics and complications by anaemia status

| Characteristics | Total N (%) | Maternal haemoglobin, N (%) | | | |
|---|---|---|---|---|---|
| | | Normal ≥110 g/L | Mild 100–109 g/L | Moderate 70–99 g/L | Severe <70 g/L |
| Deliveries | 34 087 | 28 515 (83.7) | 3703 (10.9) | 1769 (5.2) | 100 (0.3) |
| Maternal age (years) | | | | | |
| <35 | 26 433 (77.5) | 22 082 (77.4) | 2890 (78.0) | 1382 (78.1) | 79 (79.0) |
| ≥35 | 7654 (22.5) | 6433 (22.6) | 813 (22.0) | 387 (21.9) | 21 (21.0) |
| Maternal BMI | | | | | |
| <18.5 | 4920 (14.4) | 3978 (14.0) | 574 (15.5) | 336 (19.0) | 32 (32.0) |
| 18.5–23.9 | 23 913 (70.2) | 20 123 (70.6) | 2547 (68.8) | 1184 (76.9) | 59 (59.0) |
| 24–27.9 | 4329 (12.7) | 3653 (12.8) | 472 (12.7) | 195 (11.0) | 9 (9.0) |
| ≥28 | 925 (2.7) | 761 (2.7) | 110 (3.0) | 54 (3.1) | 0 (0.0) |
| Residence | | | | | |
| Urban | 21 062 (61.8) | 17 672 (62.0) | 2253 (60.8) | 1082 (61.2) | 55 (55.0) |
| Rural | 13 025 (38.2) | 10 843 (38.0) | 1450 (39.2) | 687 (38.8) | 45 (45.0) |
| Education level | | | | | |
| Junior high school or below | 2547 (7.5) | 1705 (6.0) | 516 (13.9) | 299 (16.9) | 27 (27.0) |
| Senior middle school | 9684 (28.4) | 7842 (27.5) | 1218 (32.9) | 594 (33.6) | 30 (30.0) |
| College | 15 588 (45.7) | 13 618 (47.8) | 1346 (36.3) | 586 (33.1) | 38 (38.0) |
| Master or above | 6268 (18.4) | 5350 (18.8) | 623 (18.8) | 290 (16.4) | 5 (5.0) |
| Ethnicity | | | | | |
| Han | 33 639 (98.7) | 28 129 (98.6) | 3686 (99.5) | 1730 (97.8) | 94 (94.0) |
| Minority | 448 (1.3) | 386 (1.4) | 17 (0.5) | 39 (2.2) | 6 (6.0) |
| Parity | | | | | |
| 0 | 10 488 (30.8) | 9274 (32.5) | 813 (22.0) | 369 (20.9) | 32 (32.0) |
| 1–3 | 20 189 (59.2) | 16 601 (58.2) | 2362 (62.8) | 1168 (66.0) | 58 (58.0) |
| ≥4 | 3410 (10.0) | 2640 (9.3) | 528 (14.3) | 232 (13.1) | 10 (10.0) |
| Per caput monthly family income | | | | | |
| ≤¥2500 | 5875 (17.2) | 4916 (17.2) | 622 (16.8) | 322 (18.2) | 15 (15.0) |
| ¥2500 to ¥5000 | 18 206 (53.4) | 15 225 (53.4) | 1980 (53.5) | 944 (53.4) | 57 (57.0) |
| >¥5000 | 10 006 (29.4) | 8374 (29.4) | 1101 (29.7) | 503 (28.4) | 28 (28.0) |
| Folic acid use* | 32 540 (95.5) | 27 272 (95.6) | 3490 (94.2) | 1687 (95.4) | 91 (91.0) |
| Gestational diabetes mellitus | 5424 (15.9) | 4689 (16.4) | 485 (13.1) | 250 (14.1) | 0 (0.0) |
| Gestational hypertension | 1246 (3.7) | 1012 (3.5) | 161 (4.3) | 68 (3.8) | 5 (5.0) |
| Hyperlipidaemia | 375 (1.1) | 312 (1.1) | 35 (0.9) | 28 (1.6) | 0 (0.0) |
| Hyperthyroidism | 636 (1.9) | 542 (1.9) | 71 (1.9) | 23 (1.3) | 0 (0.0) |
| Placenta previa | 606 (1.8) | 492 (1.7) | 70 (1.9) | 42 (2.4) | 2 (2.0) |

*Folic acid use before or during pregnancy.
BMI, body mass index.

face-to-face interviews. In addition, we gathered data using the database of the Electronic Maternal and Child Health Information System which recorded clinical and biochemical samples from mothers and infants during registration to delivery. The information on maternal complications during pregnancy and pregnancy outcomes of the offspring were obtained from medical records.

**Exposures and covariates**

The primary exposure was maternal Hb concentration in the first trimester (<14 weeks). Pregnant women accepted their first antenatal care at 8–14 gestational weeks and were concurrently collected 5 mL blood samples following standard operating procedures to measure the Hb concentration via a Mindray haematology analyser available at the Central Laboratory of the Hunan Provincial

Maternal and Child Health Care Hospital. Participants whose Hb level during early pregnancy was <110 g/L were considered as anaemia according to the WHO definitions.[18] We further categorised the anaemia for pregnant women into three subgroups (mild anaemia: 10.0≤Hb<110 g/L; moderate anaemia: 70≤Hb<100 g/L; severe anaemia: Hb<70 g/L).

Information on covariates included socio-demographic characteristics and maternal complications during pregnancy. The covariates for mothers were selected based on literature review as followings: maternal age at pregnancy onset (*<35 years old and ≥35 years*), body mass index (BMI) before pregnancy (*<18.5 kg/m², 18.5–23.9 kg/ m², 24–27.9 kg/m² and ≥28 kg/m2*), residence (*urban and rural*), education level (*junior high school or below, senior middle school, college and master or above*), ethnicity (*Han and Minority*), parity (*0, 1–3 and ≥4*), folic acid use before or during pregnancy (*yes or no,*), per caput monthly family income (*≤¥2500, ¥2500 to 5000, >¥5000*), gestational diabetes mellitus (*yes or no*), gestational hypertension (*yes or no*), hyperlipidaemia (*yes or no*), hyperthyroidism (*yes or no*), placenta previa (*yes or no*).[19–22] Folic acid use before or during pregnancy is defined as taking at least 0.4 mg of folic acid daily for more than 5 days per week continuously before conception or throughout the first trimester of pregnancy.[23] In China, a Maternal and Child Health Manual will be offered to every pregnant woman and recorded the following information, including the basic demographic characteristics, clinical examination and folic acid supplementation. After finishing the questionnaire, the information in the manual will be compared with the questionnaire to confirm the accuracy of gathered information.

## Outcomes

In this study, the neonatal adverse outcomes of interest were preterm birth, low birth weight, SGA and congenital malformation. Preterm birth was defined as delivery before 37 weeks of gestation. Low birth weight was defined as birth weight <2500 g. SGA was defined as birth weight below the 10th percentile of mean weight corrected or fetal sex and gestational age. We categorised congenital malformations based on ICD-10 (International Classification of Diseases 10th revision).

## Statistical analysis

This study was reported following the Strengthening the Reporting of Observational Studies in Epidemiology statement. Absolute numbers or percentages were used to describe the distributions of maternal characteristics and maternal complications during pregnancy by the anaemia status. We estimated the rate of each neonatal adverse outcome in women without anaemia and those with mild, moderate and severe anaemia. We did Mantel-Haenszel $\chi^2$ test for trends for each neonatal outcome to compare trends in the rate of neonatal outcomes by different maternal anaemia statuses. Binary logistic regression model was used to analyse the associations between neonatal outcomes and women with varying statuses of anaemia during early pregnancy. Restricted cubic spline regression was conducted to address the potential non-linearity of the association between maternal Hb levels and neonatal outcomes. Adjusted ORs (aOR) and 95% CIs were estimated by multivariate logistic regression to control for potential confounders. Restricted cubic spline regression also controlled the confounding factors. Sensitivity analyses were conducted by adjusting for different covariates in multivariate models. In model A, we adjusted for maternal age, maternal BMI, residence, education level, ethnicity, parity, folate use and per caput monthly family income. In model B, we additionally adjusted for complications during pregnancy including gestational diabetes mellitus, gestational hypertension, hyperlipidaemia, hyperthyroidism and placenta previa. EpiData V.3.1 (EpiData Association, Odense, Denmark) was used to develop the database by double entry to ensure the accuracy of the data. All analyses were performed using R software (V.3.5.0). All tests were two-tailed, with p<0.05 set as the statistically significant difference.

## RESULTS

### Characteristics of participants

We included 34 087 pregnant women in our study. A total of 5572 (16.3%) participants were diagnosed with anaemia in the first trimester. The prevalence of mild anaemia, moderate anaemia and severe anaemia was 10.9%, 5.2% and 0.3%, respectively. The participants were concentrated in the age group of <35 years (77.5%). 70.2% of the participants had a pre-pregnancy BMI of 18.5–23.9 kg/m², and most of the individuals lived in urban (61.8%). Their educational level was mostly senior middle school (28.4%) and college (45.7%). Of these individuals, 98.7% were of Han ethnicity, and 1.3% were members of other ethnic groups (includes >50 ethnic groups). Of the 34 087 pregnant women recruited in our study, a total of 15.9% of the participants reported gestational diabetes mellitus, 3.7% had gestational hypertension and 1.8% reported with placenta previa. Distribution of maternal characteristics and complications by anaemia statuses were displayed in table 1.

### Prevalence of neonatal adverse outcomes across maternal anaemia status

The prevalence of preterm birth varied by maternal anaemia status (table 2), ranging from 10.9% among women without anaemia to 40.0% among women with severe anaemia (P$_{trend}$<0.001). Likewise, the prevalence of low birth weight and SGA infants raised from 7.6% to 44.0% and 14.1% to 35.0% among pregnant women with more severe anaemia compared with no anaemia, respectively (P$_{trend}$<0.001). There were positive relationships between the rate of preterm birth, low birth weight as well as SGA and the severity of maternal anaemia. Whereas the statistically significant associations between the rate

**Table 2** Prevalence (% and 95% CI) of neonatal adverse outcomes according to maternal anaemia status

| Maternal anaemia | Preterm birth % (95% CI) | Low birth weight % (95% CI) | SGA % (95% CI) | Congenital malformation % (95% CI) |
|---|---|---|---|---|
| Normal (≥110 g/L) | 10.9 (10.5 to 11.2) | 7.6 (7.2 to 7.9) | 14.1 (13.7 to 14.5) | 4.4 (4.2 to 4.7) |
| Mild (100–109 g/L) | 15.7 (14.5 to 16.9) | 13.7 (12.6 to 14.8) | 20.2 (18.9 to 21.5) | 3.9 (3.3 to 4.6) |
| Moderate (70–99 g/L) | 18.1 (16.3 to 19.9) | 17.8 (16.0 to 19.6) | 22.8 (20.8 to 24.7) | 5.0 (4.0 to 6.0) |
| Severe (<70 g/L) | 40.0 (30.2 to 49.8) | 44.0 (34.1 to 53.9) | 35.0 (25.5 to 44.5) | 4.0 (0.1 to 7.9) |
| $P_{trend}$ | <0.001 | <0.001 | <0.001 | 0.899 |

SGA, small for gestational age.

of congenital malformation and different maternal anaemia statuses were not observed ($P_{trend}$>0.05).

## Association of maternal anaemia with the risk of neonatal outcomes

To assess the association between the severity of maternal anaemia and adverse neonatal outcomes, we compared the risk of the neonatal outcomes for each anaemia status to the risk of neonatal outcomes in women without anaemia (Hb≥110 g/L). After adjusting for potential confounders including age, BMI, residence, education level, ethnicity, parity, folic acid use and per caput monthly family income (table 3, model A), the risk estimates of preterm birth (mild anaemia aOR 1.37 (95% CI 1.25 to 1.52), moderate anaemia aOR 1.54 (95% CI 1.35 to 1.76) and severe anaemia aOR 4.03 (95% CI 2.67 to 6.08), respectively), low birth weight (mild anaemia aOR 1.61 (95% CI 1.44 to 1.79), moderate anaemia aOR 2.01 (95% CI 1.75 to 2.30) and severe anaemia aOR 6.11 (95% CI 3.99 to 9.36), respectively) and SGA (mild anaemia aOR 1.37 (95% CI 1.25 to 1.52), moderate anaemia aOR 1.54 (95% CI 1.35 to 1.76) and severe anaemia aOR 2.61 (95% CI 1.74 to 4.50), respectively) significantly increased in offspring of pregnant women who had anaemia. However, we did not find that women with anaemia were associated with the risk of congenital malformation in offspring (mild anaemia aOR 0.89 (95% CI 0.75 to 1.07), moderate anaemia aOR 1.17 (95% CI 0.93 to 1.47) and severe anaemia aOR 0.86 (95% CI 0.32 to 2.36), respectively). Figure 2 exhibited the restricted cubic spline for exploring the non-linear association between maternal Hb levels and the risk of neonatal outcomes after adjusting model A. Maternal Hb concentrations displayed the U-shaped relationship curve with the likelihood of preterm birth, low birth weight and SGA in offspring (all values $P_{non-linear}$<0.05). The significantly non-linear association between maternal Hb levels and the risk of congenital malformation in offspring was not observed ($P_{non-linear}$>0.05). The similar findings were illustrated in figure 3 after adjusting model B. Adjusting for different covariates did not substantially affect the risk estimates of adverse neonatal outcomes, which were similar for model A and model B. AORs and 95% CIs for covariates in model A and B associated with neonatal

adverse outcomes were illustrated in online supplemental table 1.

## DISCUSSION

We conducted a prospective cohort study using information about pregnant women in Hunan Provincial Maternal and Child Health Care Hospital, China. The prevalence of gestational anaemia was 16.3% in the present study which was comparable with results previously reported in China.[24 25] After adjusting for maternal confounders, maternal anaemia during early pregnancy was associated with the risk of preterm birth, low birth weight and SGA. We also observed non-linear associations of maternal Hb concentrations with the risk of preterm birth, low birth weight and SGA in offspring. No association was found between anaemia in early pregnancy and the risk of congenital malformations, nor was a non-linear association found between Hb levels in early pregnancy and the risk of congenital malformations. Sensitivity analysis verified the stability of the results. Furthermore, the positive association of the rate of preterm birth, low birth weight and SGA with the severity of maternal anaemia in the first trimester was identified.

Published literature had already reported associations of anaemia during pregnancy with neonatal outcomes such as preterm birth, low birth weight and SGA. However, these findings had not been consistent.[26–28] This might be that various researches defined anaemia during pregnancy with different cut-off thresholds. Besides, the associations between maternal anaemia and neonatal outcomes varied depending on the timing of the maternal Hb measurement.[27] In the current study, we specified that Hb was measured in the first trimester to analyse the associations between anaemia during early pregnancy and neonatal outcomes. Our data showed that maternal anaemia during early pregnancy had negative effects on preterm birth, low birth weight and SGA. Consistent with our findings, a meta-analysis reported an increased risk of low birth weight among women with lower Hb concentrations during overall or any stage of pregnancy.[8] Associations between maternal anaemia

**Table 3** Adjusted risk ratios for neonatal adverse outcomes associated with severity of anaemia during early pregnancy†

| Outcome | Model A‡ | | | | Model B§ | | | |
|---|---|---|---|---|---|---|---|---|
| | Mild 100–109 g/L | Moderate 70–99 g/L | Severe <70 g/L | Overall <110 g/L | Mild 100–109 g/L | Moderate 70–99 g/L | Severe <70 g/L | Overall <110 g/L |
| Preterm birth | 1.37 (1.25 to 1.52)* | 1.54 (1.35 to 1.76)* | 4.03 (2.67 to 6.08)* | 1.47 (1.35 to 1.59)* | 1.37 (1.24 to 1.51)* | 1.55 (1.36 to 1.77)* | 4.05 (2.68 to 6.12)* | 1.46 (1.35 to 1.59)* |
| Low birth weight | 1.61 (1.44 to 1.79)* | 2.01 (1.75 to 2.30)* | 6.11 (3.99 to 9.36)* | 1.80 (1.64 to 1.96)* | 1.60 (1.43 to 1.78)* | 2.02 (1.76 to 2.32)* | 6.25 (4.06 to 9.61)* | 1.79 (1.64 to 1.96)* |
| SGA | 1.37 (1.25 to 1.52)* | 1.54 (1.35 to 1.76)* | 2.61 (1.74 to 4.50)* | 1.47 (1.35 to 1.59)* | 1.37 (1.24 to 1.51)* | 1.55 (1.36 to 1.77)* | 2.60 (1.73 to 4.49)* | 1.46 (1.35 to 1.59)* |
| Congenital malformation | 0.89 (0.75 to 1.07) | 1.17 (0.93 to 1.47) | 0.86 (0.32 to 2.36) | 0.98 (0.85 to 1.13) | 0.88 (0.74 to 1.06) | 1.15 (0.92 to 1.44) | 0.83 (0.30 to 2.28) | 0.96 (0.83 to 1.12) |

*P value < 0.005.

†Non-anaemic pregnant women as reference.

‡Model A adjusted for maternal age, maternal body mass index, residence, education level, ethnicity, parity, folate use, and per caput monthly family income.

§Model B adjusted for all maternal complications during pregnancy in addition to covariates in model A.

SGA, small for gestational age.

in the first trimester and neonatal outcomes were inconclusive. According to the results of Ronkainen et al,[28] the risk of SGA increased in the offspring of pregnant women whose level of Hb was <11.0 g/dL in the first trimester. But this study showed that no association between low maternal Hb level and preterm birth. A study performed in the UK came to the same conclusion that low maternal Hb during early pregnancy is associated with SGA, but not preterm birth.[29] However, Ren's result implied that rates of low birth weight, preterm birth and SGA increased with the decrease in first-trimester Hb concentration in this Chinese population which was consistent with our research results.[30] Additionally, our study discovered that maternal Hb concentrations displayed a U-shaped relationship with the risk of preterm birth, low birth weight and SGA in offspring. Consistent with our findings, a study conducted in the UK reported a U-shaped association between maternal Hb values and the risk of low birth weight and preterm birth.[31] A study performed in Chinese pregnant women indicated that U-shaped relationships between first-trimester Hb concentrations and the risk of preterm birth, low birth weight and SGA were observed without adjusting for confounding factors.[32]

Hb carries oxygen to the placenta, which is essential for normal fetal growth in the uterus. Anaemia or low Hb levels may stimulate changes in placental angiogenesis and affect placental transit function.[33] As a result, it reduces the oxygen supply and nutrient supplementation to the fetus, resulting in intrauterine growth retardation and low birth weight.[34] Increased placental vascularisation in women with low Hb levels suggests earlier placental maturation, which may contribute to an increased risk of preterm birth.[34] Our study discovered that pregnant women with underweight were associated with severe anaemia. An Indian cohort study suggested that over third anaemic pregnant women coexisted with underweight, and were associated with the increased risk of low birth weight in offspring,[35] implying that maternal anaemia possibly caused by malnutrition statuses had an impact on intrauterine fetal nutrition and led to low birth weight in the offspring. In addition, low Hb may lead to chronic hypoxia which in turn may cause a stress response of increased placental corticotropin-releasing hormone secretion, inducing adverse consequences such as preterm birth, gestational hypertension and so forth.[36] Nevertheless, the underlying biological mechanisms involved in the associations between maternal anaemia and preterm birth, low birth weight as well as SGA remain to be further elucidated.

Organogenesis occurs in the first trimester of pregnancy, during which the primary structures of most organs develop such as the heart, lung, liver and so on. In the early stage of development, the embryo and the fetus undergo intense epigenetic reprogramming.[14 15]

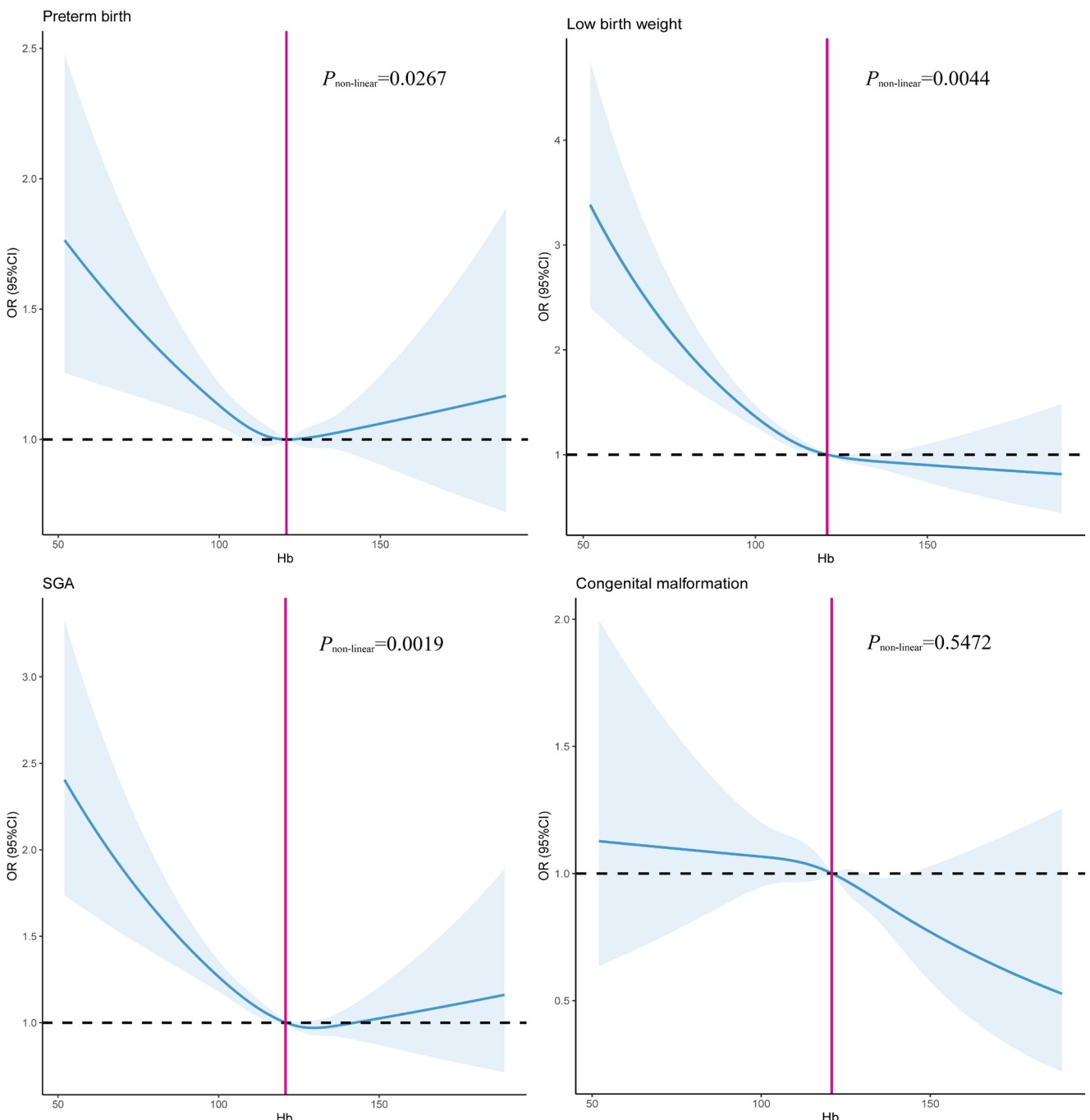

**Figure 2** Restricted cubic spline of the association between maternal haemoglobin levels and the risk of neonatal outcomes after adjusting model A. Model A adjusted for maternal age, maternal body mass index, residence, education level, ethnicity, parity, folate use and per caput monthly family income. SGA, small for gestational age.

Previous studies have shown that epigenetic changes might play a significant role in offspring diseases caused by gestational complications.[16] Gestational anaemia is one of the most common gestational complications. Our study revealed a lack of association between maternal anaemia during early pregnancy and congenital malformation. Shi *et al*[37] also found that no association between low birth weight and maternal anaemia during overall pregnancy.

Significantly, our findings suggested that pregnant women with severe anaemia had a lower prevalence of gestational diabetes mellitus and Lin *et al*[38] also provided evidence that the prevalence of gestational diabetes mellitus decreased in women with anaemia. Additionally, Lao and Ho[39] discovered the association of iron deficiency anaemia with the decreased prevalence of gestational diabetes mellitus. Conversely, Liu *et al*[40] yielded a different result that there was no significant association

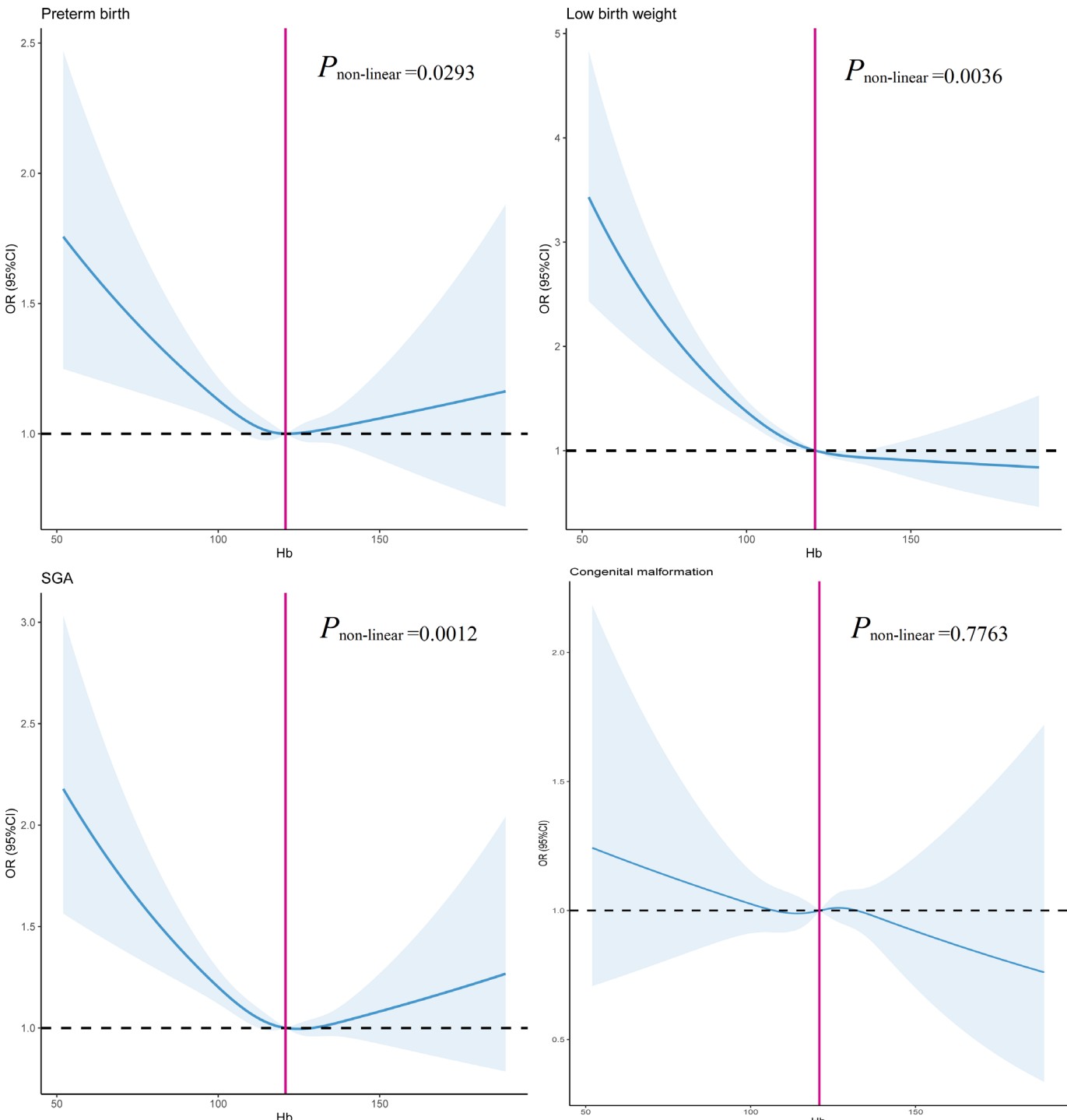

**Figure 3** Restricted cubic spline of the association between maternal haemoglobin levels and the risk of neonatal outcomes after adjusting model B. Model B adjusted for all maternal complications during pregnancy in addition to covariates in model A. SGA, small for gestational age.

between iron deficiency anaemia status and gestational diabetes mellitus (GDM) risk. Therefore, the association between anaemia and gestational diabetes mellitus remains to be further explored.

This study used a large sample of pregnant women and took place in a prospective cohort study which minimised recall bias and ensured the reliability of the study. Additionally, our study specified that Hb was measured during early pregnancy to analyse the association between anaemia during early pregnancy and neonatal outcomes. However, some limitations should be considered. First, the study objects were all from the same hospital, and the sample sources may be concentrated in a certain group of population. This may affect the representativeness of the sample and thus affect extrapolation. Second, we did not collect information on maternal anaemia remedies

and Hb concentrations in childbirth. Third, there was no assurance that the results would not be impacted by residual confounding factors even after we adjusted for a large number of potential confounding factors.

## CONCLUSION

In conclusion, it is of concern that anaemia during early pregnancy regardless of the severity may significantly increase the risk of neonatal outcomes such as preterm birth, low birth weight and SGA. According to the results of our study, we suggest that the rate of neonatal outcomes may increase with the severity of maternal anaemia. However, more prospective studies with larger samples are needed to verify the authenticity of the results and explore the underlying mechanisms. Pre-pregnancy screening for anaemia and implementation of interventions are necessary to ensure adequate Hb concentrations during pregnancy and to reduce the adverse effects of maternal anaemia on the health of their offspring.

**Author affiliations**
[1] Department of Epidemiology and Health Statistics, Xiangya School of Public Health, Central South University, Changsha, Hunan, China
[2] Department of Science and Education, Xiangya Changde Hospital, Changde, China
[3] Public Health Institute, Changsha Medical University, Changsha, Hunan, China
[4] The Hospital of Trade-Business in Hunan Province, Changsha, Hunan, China
[5] Hunan Provincial Maternal and Child Health Care Hospital, Changsha, Hunan, China

**Acknowledgements** The authors would like to thank the editors and reviewers for their suggestions and all colleagues working in the Maternal and Child Health Promotion and Birth Defect Prevention Group.

**Contributors** YL and TZ wrote the main manuscript text. YC and TZ analysed the data statistical analyses. JQ, YC and YL reviewed and revised the manuscript. XL, YL, XS, SZ, MS, JW and JS have collected the data. JQ was the guarantor. All authors have read and approved the final version of the manuscript.

**Funding** This work was supported by the Project Funded by National Natural Science Foundation Program of China (82073653 and 81803313), Hunan Outstanding Youth Fund Project (2022JJ10087), Hunan Provincial Science and Technology Talent Support Project (2020TJ-N07), Hunan Provincial Key Research and Development Program (2018SK2063), Open Project from NHC Key Laboratory of Birth Defect for Research and Prevention (KF2020006), Natural Science Foundation of Hunan Province (2018JJ2551) and Science and Technology Planning Project of Guangdong Province (2020A1414010152).

**Competing interests** None.

**Patient and public involvement** Patients and/or the public were not involved in the design, or conduct, or reporting, or dissemination plans of this research.

**Patient consent for publication** Not applicable.

**Ethics approval** This study complied with the ethical principles of the Declaration of Helsinki. Ethical approval was given by the Ethics Committee of Xiangya School of Public Health, Central South University (No. XYGW-2018-36). Participants gave informed consent to participate in the study before taking part.

**Provenance and peer review** Not commissioned; externally peer reviewed.

**Data availability statement** Data are available upon reasonable request. The data sets used and/or analysed during the current study are available from the corresponding author on reasonable request.

**ORCID iD**
Jiabi Qin http://orcid.org/0000-0002-9360-4991

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
