## [Reviewer comments · BMJ Paediatrics Open]

ARTICLE DETAILS

TITLE (PROVISIONAL)	Maternal anemia during early pregnancy and the risk of neonatal outcomes: a prospective cohort study in Central China
AUTHORS	Chen, Yige Zhong, Taowei Song, Xinli Zhang, Senmao Sun, Mengting Liu, Xiaoying Wei, Jianhui Shu, Jing Liu, Yiping Qin, Jiabi

VERSION 1 - REVIEW

REVIEWER	Dr. Reeta Bora Assam Medical College, Pediatrics
REVIEW RETURNED	13-Mar-2023

GENERAL COMMENTS	Thanks for giving me the opportunity to review the manuscript. Relevance of topic: The relation of maternal anemia with the neonatal outcomes of Small for gestational age, prematurity, and low birth weight has been documented by a number of previous studies. The only novelty of the present study is correlating maternal 1st-trimester anemia with neonatal outcome Introduction: The background and need for the study have been explained well. Abstract: 1. Authors have mentioned the total number of pregnant women included in the study to be 3,408 whereas later in the manuscript has been mentioned as 34,087. Correction is needed.2. The references all throughout the manuscript have been numbered after a full stop, and need to be put before the full stop. Methodology: 1. The authors have mentioned it as a prospective cohort study with data collected from 13th March'2013 til 31st January'20. However, the study registration has been dated 14th June'2018, indicating registration is being done retrospectively.2. Why maternal Hemoglobin estimation in 1st trimester has been mentioned by authors as exposure is not clear.3. Authors have mentioned that the "time sequence of maternal complications to the onset of anemia was unknown but may have been associated with outcomes." The study is mentioned as a prospective study, the information should have been ideally
---

	available to authors. 4. The unit used to describe Hb level should be g/L rather than g/dl (Ex 110 g/L rather than 11.0 g/dl) Results:  1. From the text it is not clear whether 15.9% of pregnant women who had GDM, 1.8% of mothers having placenta previa and 3.7% having PIH belonged to the total population of 34087 pregnant women or 5572 pregnant women having anemia. However, it is shown in the table. This statement needs clarification. 2. Did the anemic mothers receive any specific management, like iron injection, especially the group having severe anemia (<70g/L)? Was there any change in Hb level by the time of delivery from the 1st-trimester Hb level or did the mothers remain anemic? This information is important to get convinced that 1st-trimester maternal Hb level influences the neonatal outcome. 3. Figure 1: Flow chart showing the process of participant recruitment is missing Discussion Amongst the severely anemic mothers highest percentage (32%) had BMI <23 and only 9% were overweight with BMI 23-28, while in the non-anemic group, only 14% had BMI <28, and 14.5% of mothers of this same group had BMI of indicating overweight (23-28)and obesity(>28). This observation needs to be pointed out in the discussion as it indicates maternal poor nutrition as a cause of maternal anemia. The tables also show that GDM was observed in 16.4% of non-anemic mothers, 13.1% of mild and 14.1% of moderately anemic mothers while none in severely anemic mothers
--	---

REVIEWER	Dr. Guochong Chen Soochow University Medical College, Guo-Chong Chen
REVIEW RETURNED	13-Mar-2023

GENERAL COMMENTS	Thanks for inviting me to review this manuscript entitled “Maternal anemia during early pregnancy and the risk of neonatal outcomes: a prospective cohort study in Central China”. Overall, I feel this study interesting and it evaluates the association of maternal anemia during early pregnancy and the risk of neonatal outcomes in an understudied population. There are a few points which need to be clarified further:  1. There are many factors in pregnancy that can affect neonatal outcomes. How can the authors exclude the impacts of other risk factors when evaluating the relationship between maternal anemia and neonatal outcome? 2. The authors need to provide more information on why they chose the first-trimester period. 3. What is the definition of folic acid use? Any classification criteria for folic acid use? (Page 5/15, line 33-34) 4. Were the participants asked about the level of income? 5. The authors stated: “After recruitment, the specially trained investigators used a structured questionnaire to collect the information on maternal sociodemographic characteristics through face-to-face interview.” What’s about recall bias?
---

REVIEWER	Dr. Peter Flom Peter Flom Consulting
REVIEW RETURNED	21-Mar-2023

GENERAL COMMENTS	I confine my remarks to statistical and methodological aspects of this paper. Logistic regression is correct, but there are some fairly large changes to be made before I can recommend publication. p. 2 Line 6 and elsewhere "Mutlivariate" should be "multiple". Technically, multivariate logistic regression would mean more than one dependent variable. Line 10 There seems to be a typo here in "34,08" Line 12 "upward curve" should be "positive" Lines 14-20 The aORs for each outcome are identical. This seems like a typo, as well. Also, the relationship here is not positive but negative. (same thing on p. 7). Line 20-21 This is badly worded. The association *was* observed. It's just that it was not statistically significant. Please give the aOR and CI, even though it wasn't significant. p. 3 line 9-10 This is the opposite of what was reported in the abstract p. 5 Categorizing anemia and other continuous independent variables is not a good idea. Leave Hb concentration, age, and so on continuous. You can also use splines of any continuous independent variables. If you want to present results at specific levels of Hb, that's fine, but the analysis should be done on the continuous Hb. See my paper: https://medium.com/@peterflom/what-happens-when-we-categorize-an-independent-variable-in-regression-77d4c5862b6c The numbers in table 2 and table 3 show that severe was much worse than the mild or moderate. This is another reason to use Hb as a continuous measure and use a spline of this to look for nonlinearities. p. 6 Why use both CMH and binary logistic regression? Here, logistic reg. seems more appropriate. Also, if you follow my advice and leave Hb continuous, the CMH makes no sense. Table 3 It would be good to show the aORs for the covariates as well as for Hb. This isn't absolutely necessary, but it would make the table more informative.
---

VERSION 1 – AUTHOR RESPONSE

Reviewer 1

Comment 1: Abstract:

Authors have mentioned the total number of pregnant women included in the study to be 3,408 whereas later in the manuscript has been mentioned as 34,087. Correction is needed.

Response: Thank you very much for your careful review. We have corrected it. (Page 2, Line 10)

Comment 2: Abstract:

The references all throughout the manuscript have been numbered after a full stop, and need to be put before the full stop.

Response: Thank you very much for your careful review. We have corrected the format.

Comment 3: Methodology:

The authors have mentioned it as a prospective cohort study with data collected from 13th March'2013 til 31st January'20. However, the study registration has been date d, indicating registration is being done retrospectively.

Response 3: Thank you for your careful review. We apologize for your misunderstanding caused by our unclear presentation of the manuscript. In fact, this study was designed as an observational registry and did not involve clinical trials, and it is not mandatory for registration to be completed prior to the inclusion of the first participant. Additionally, the protocol had been approved by the ethics committee before we recruited pregnant women and written informed consent was also obtained from all participants. The registration date is not a criterion to judge the type of study design. The mean and standard deviation of maternal age for those included before and after June 14, 2018 were 31.12 ± 4.51 and 31.86 ± 4.48 years, respectively, and the prevalence of maternal anemia for those included before and after June 14, 2018 were 15.9% and 16.5%, respectively. There is no heterogeneity here. Therefore, the date of registration also does not affect the study results. We thank you again for your careful review.

Comment 4: Methodology

Why maternal Hemoglobin estimation in 1st trimester has been mentioned by authors as exposure is not clear.

Response 4: Thank you for your careful review. As we described, this study was designed as a prospective cohort study. We recruited pregnant women who received their first antenatal care during 8-14 gestation weeks and intended to get prenatal care consistently throughout gestation at Hunan Provincial Maternal and Child Health Care Hospital, in Hunan Province, China. At enrollment, we collected maternal blood samples to measure the Hb concentration and neonatal outcomes was gathered using clinical record at the time of delivery. We also used a structured questionnaire to collect information on maternal sociodemographic characteristics. As we mentioned, this study aimed to assess the association between anemia during early pregnancy and the risk of neonatal outcomes, and then we conducted regression models to estimate the effect of anemia for the risk of neonatal outcomes. Most importantly, the chronological relationship between anemia and neonatal outcome was explicit. Therefore, maternal anemia was defined as exposure in the present study.

Comment 5: Methodology

Authors have mentioned that the “time sequence of maternal complications to the onset of anemia was unknown but may have been associated with outcomes.” The study is mentioned as a prospective study, the information should have been ideally available to authors.

Response 5: Thank you for your detailed review. We completely agree with the reviewer that the information on time sequence of maternal complications to the onset of anemia should have been ideally available to us. We have deleted the relevant expression in Exposures and covariates and Discussion section.

Comment 6: Methodology

The unit used to describe Hb level should be g/L rather than g/dl (Ex 110 g/L rather than 11.0 g/dl)

Response 6: Thank you very much for your careful review. We have revised it.

Comment 7: Results

From the text it is not clear whether 15.9% of pregnant women who had GDM, 1.8% of mothers having placenta previa and 3.7% having PIH belonged to the total population of 34087 pregnant women or 5572 pregnant women having anemia. However, it is shown in the table. This statement needs clarification.

Response 7: Thank you for your kind suggestion. We have clarified it as follows: “Of the 34,087 pregnant women recruited in our study, a total of 15.9% of the participants reported gestational diabetes mellitus, 3.7% had gestational hypertension, and 1.8% reported with placenta previa.” (Page 6-7, Line 42-44, 1-2)

Comment 8: Results

Did the anemic mothers receive any specific management, like iron injection, especially the group having severe anemia (<70g/L)? Was there any change in Hb level by the time of delivery from the 1st-trimester Hb level or did the mothers remain anemic? This information is important to get convinced that 1st-trimester maternal Hb level influences the neonatal outcome.

Response 8: Thank you very much for your careful review and constructive suggestions about our manuscript. We completely agree with the reviewer that prenatal management of anemia, such as iron injection, influenced the associations of maternal anemia during early pregnancy with the risk of neonatal outcomes. The duration of pregnancy is split into three trimesters: the first trimester (< 14 weeks), the second trimester (14–27+6 weeks), and the third trimester (\geq 28 weeks). Organogenesis takes place in the first trimester of pregnancy, during which the majority of organs develop their preliminary structure[1, 2]. In this early stage, embryos and fetuses are liable to intense epigenetic reprogramming, through which gestational complications, intrauterine environments and an inadequate nutritional status spur on neonatal disorders[3, 4]. As one of the most common gestational complications, the environment of nutritional deficiency and hypoxia caused by anemia during early pregnancy also may lead to the occurrence of adverse neonatal outcomes[5]. In addition, anemia during the first trimester was more detrimental[6]. Overall, anemia in the first trimester of pregnancy is more likely to lead to adverse neonatal outcomes and is more harmful than anemia in other trimesters. Therefore, this study is mainly concerned with the connection between maternal anemia during early pregnancy and the risk of neonatal outcomes. We must admit that the investigation of prenatal anemia management and Hb levels at delivery might provide additional crucial insights into this subject, and thus we added the shortcoming to the limitation section in the Discussion.

“Secondly, we did not collect information on maternal anemia remedies and Hb concentrations in childbirth.” (Page 8, Line 27-28)

Results

Figure 1: Flow chart showing the process of participant recruitment is missing

Response 9: Thank you very much for your careful review. We have added it.

Comment 10: Discussion

Amongst the severely anemic mothers highest percentage (32%) had BMI <23 and only 9% were overweight with BMI 23-28, while in the non-anemic group, only 14% had BMI <28, and 14.5% of mothers of this same group had BMI of indicating overweight (23-28) and obesity (>28). This observation needs to be pointed out in the discussion as it indicates maternal poor nutrition as a cause of maternal anemia. The tables also show that GDM was observed in 16.4% of non-anemic mothers, 13.1% of mild and 14.1% of moderately anemic mothers while none in severely anemic mothers

Response 10: We sincerely thank you for this comment and constructive suggestions. We have discussed this subject in Discussion as follows: “Our study discovered that pregnant females with underweight were associated with severe anemia. An Indian cohort study suggested that over third anemic pregnant women coexisted with underweight, and were associated with the increased risk of low birth weight in offspring[35], implying that maternal anemia possibly caused by malnutrition statuses had an impact on intrauterine fetal nutrition and led to low birth weight in the offspring.” (Page 8, Line 37-42)

“Significantly, our findings suggested that pregnant females with severe anemia had a lower prevalence of gestational diabetes mellitus and Lin et al.[38] also provided evidence that the prevalence of gestational diabetes mellitus decreased in women with anemia. Additionally, Lao et al.[39] discovered the association of iron deficiency anemia with the decreased prevalence of gestational diabetes mellitus. Conversely, Liu et al.[40] yielded a different result that there was no significant association between iron deficiency anemia status and GDM risk. Therefore, the association between anemia and gestational diabetes mellitus remains to be further explored.” (Page 9, Line 12-18)

Reviewer 2:

Comment 1: There are many factors in pregnancy that can affect neonatal outcomes. How can the authors exclude the impacts of other risk factors when evaluating the relationship between maternal anemia and neonatal outcome?

Response 1: We sincerely thank you for this comment. We totally agree with the reviewer that neonatal outcomes could be influenced by several factors in pregnancy. To minimize the bias caused by confounding factors, we have controlled these factors in pregnancy known and suspected to be associated with neonatal outcomes, however, there are still confounding factors that cannot be taken into account despite our earnest efforts. Therefore, we have adjusted them in multivariate logistic regression to estimate adjusted odds ratios (aOR) and 95% confidence intervals (95% CIs) when examining the relationships between maternal anemia during early pregnancy and the risk of neonatal outcomes. Restricted cubic spline regression also controlled the confounding factors. Overall, the present study still had some shortcomings, which we have added to the limitation sections in the manuscript.

“Thirdly, there was no assurance that the results would not be impacted by residual confounding factors even after we adjusted for a large number of potential confounding factors.” (Page 9, Line 28-30)

Comment 2: The authors need to provide more information on why they chose the first-trimester period.

Response 2: Thank you very much for your careful review. As is known to all, the duration of pregnancy is split into three trimesters: the first trimester (< 14 weeks), the second trimester (14–27+6 weeks), and the third trimester (≥ 28 weeks). Organogenesis takes place in the first trimester of pregnancy, during which the majority of organs develop their preliminary structure[1, 2]. In this early stage, embryos and fetuses are liable to intense epigenetic reprogramming, through which an intrauterine environment and an inadequate nutritional status spur on neonatal disorders[3, 4]. As one of the most common gestational complications, the nutritional deficiency status and hypoxia caused by anemia during early pregnancy also may lead to the occurrence of adverse neonatal outcomes[5]. In addition, anemia during the first trimester was more detrimental[6]. Overall, anemia in the first trimester of pregnancy is more likely to lead to adverse neonatal outcomes and is more harmful than anemia in other trimesters. Therefore, this study is mainly concerned with the connection between maternal anemia during early pregnancy with the risk of neonatal outcomes. We have also added it to the revised manuscript.

“In addition, organogenesis occurs in first-trimester pregnancy, during which embryos and fetuses are liable to intense epigenetic reprogramming. It has been suggested that gestational complications, intrauterine environments and a maternal inadequate nutritional status could ultimately result in a range of adverse neonatal outcomes via epigenetic change[14-16]. What’s more, anemia during the first trimester was more detrimental[17]. Overall, anemia in the first trimester of pregnancy is more likely to lead to adverse neonatal outcomes and is more harmful than anemia in other trimesters.” (Page 4, Line 25-31)

Comment 3: What is the definition of folic acid use? Any classification criteria for folic acid use? (Page 5/15, line 33-34)

Response 3: Thank you very much for your careful review. we have added the classification criteria for folic acid use in the revised manuscript.

“Folic acid use before or during pregnancy is defined as taking at least 0.4 mg of folic acid daily for more than five days per week continuously before conception or throughout the first trimester of pregnancy” (Page 5, Line 42-44)

Comment 4: Were the participants asked about the level of income?

Response 4: We sincerely thank you for this comment. For the level of income of the groups, our questionnaire included items on per caput monthly family income (RMB), and face-to-face interviews by professionally trained investigators were used to collect this information. Following the kind suggestion of this reviewer, we included per caput monthly family income (RMB) to the baseline characteristics (Table 1). Furthermore, we have recalculated and updated the adjusted OR values in Table 3 due to the addition of the new confounding factor per caput monthly family income (RMB).

Comment 5: The authors stated: “After recruitment, the specially trained investigators used a structured questionnaire to collect the information on maternal sociodemographic characteristics through face-to-face interview.” What’s about recall bias?

Response 5: Thank you very much for your careful review. In China, each pregnant woman is offered a Maternal and Child Health Manual. The following information, including the basic demographic characteristics, clinical examination, and folic acid supplementation, was recorded in the manual by

the obstetrician or the woman. After completing the questionnaire, we consulted the manual and maternal medical records to further confirm the corresponding information. We have also elaborated on it in the revised manuscript as follows: "In China, a Maternal and Child Health Manual will be offered to every pregnant woman and recorded the following information, including the basic demographic characteristics, clinical examination, and folic acid supplementation. After finishing the questionnaire, the information in the manual will be compared to the questionnaire to confirm the accuracy of gathered information." (Page 6, Line 1-4)

Reviewer 3

Comment 1: p. 2

Line 6 and elsewhere "Mutlivariate" should be "multiple". Technically, multivariate logistic regression would mean more than one dependent variable.

Response 1: Thank you very much for your careful review. We have revised it

Comment 2: p. 2

Line 10 There seems to be a typo here in "34,08"

Response 2: We thank you very much for your careful review. We have amended it.

Comment 3: p. 2

Line 12 "upward curve" should be "positive"

Response 3: Thank you very much for your careful review. We have revised it.

Comment 4: p. 2

Lines 14-20 The aORs for each outcome are identical. This seems like a typo, as well. Also, the relationship here is not positive but negative. (same thing on p. 7).

Response 4: We thank you very much for your careful review. We have revised it.

Comment 5: p. 2

Line 20-21 This is badly worded. The association *was* observed. It's just that it was not statistically significant. Please give the aOR and CI, even though it wasn't significant.

Response 5: We thank you very much for your careful review. Due to the word limit of the Abstract, we report the aOR and CI in the Results section.

"However, we did not find that women with anemia were associated with the risk of congenital malformation in offspring (mild anemia aOR 0.89 (95% CI 0.75–1.07), moderate anemia aOR 1.17 (95% CI 0.93–1.47), and severe anemia aOR 0.86 (95% CI 0.32–2.36), respectively)." (Page 7, Line 24-27)

Comment 6: p. 3 line 9-10 This is the opposite of what was reported in the abstract

Response 6: Thank you for your careful review. We apologize for your misunderstanding caused by our unclear presentation of the manuscript. We have corrected the relevant expression as follows: "There is a positive relationship between the rate of preterm birth, low birth weight as well as SGA and the severity of maternal anemia." (Page 3, Line 9-10)

Comment 7: p. 5

Categorizing anemia and other continuous independent variables is not a good idea. Leave Hb concentration, age, and so on continuous. You can also use splines of any continuous independent variables.

If you want to present results at specific levels of Hb, that's fine, but the analysis should be done on the continuous Hb.

See my paper: <https://medium.com/@peterflom/what-happens-when-we-categorize-an-independent-variable-in-regression-77d4c5862b6c>

The numbers in table 2 and table 3 show that severe was much worse than the mild or moderate. This is another reason to use Hb as a continuous measure and use a spline of this to look for nonlinearities.

Response 7: We thank you very much for your careful review and constructive suggestions. According to your comment, we have conducted the restricted cubic spline regression to explore the association of Hb concentrations as a continuous variable with neonatal outcomes. Thank you again for your constructive suggestions.

“After adjustment, the association of early pregnancy anemia and Hb levels with the the risk of preterm birth (mild anemia aOR 1.37 (95% CI 1.25-1.52), moderate anemia aOR 1.54 (95% CI 1.35-1.76), and severe anemia aOR 4.03 (95% CI 2.67-6.08), respectively), low birth weight (mild anemia aOR 1.61 (95% CI 1.44-1.79), moderate anemia aOR 2.01 (95% CI 1.75-2.30), and severe anemia aOR 6.11 (95% CI 3.99-9.36), respectively) and SGA (mild anemia aOR 1.37 (95% CI 1.25-1.52), moderate anemia aOR 1.54 (95% CI 1.35-1.76), and severe anemia aOR 2.61 (95% CI 1.74-4.50), respectively; Pnon-linear<0.05) was observed. However, no association was found between early pregnancy anemia or Hb levels and the risk of congenital malformations. Sensitivity analysis verified the stability of the results.” (Page 2, Line 13-22)

“Restricted cubic spline regression was conducted to address the potential nonlinearity of the association between maternal Hb levels and neonatal outcomes.” (Page 6, Line 20-21)

“Restricted cubic spline regression also controlled the confounding factors.” (Page 6, Line 23-24)

“Figure 2 exhibited the restricted cubic spline for exploring the non-linear association between maternal Hb levels and the risk of neonatal outcomes after adjusting model A. Maternal Hb concentrations displayed the U-shaped relationship curve with the likelihood of preterm birth, low birth weight and SGA in offspring (all values Pnon-linear<0.05). The significantly non-linear association between maternal Hb levels and the risk of congenital malformation in offspring was not observed (Pnon-linear>0.05). Similar findings were illustrated in Figure 3 after adjusting model B.” (Page 7, Line 27-33)

“We also observed non-linear associations of maternal Hb concentrations with risk of preterm birth, low birth weight and SGA in offspring. No association was found between anemia in early pregnancy and the risk of congenital malformations, nor was a non-linear association found between Hb levels in early pregnancy and the risk of congenital malformations” (Page 7, Line 43-44; Page 8, Line 1-3)

“Additionally, our study discovered that maternal Hb concentrations displayed a U-shaped relationship with the risk of preterm birth, low birth weight and SGA in offspring. Consistent with our findings, a study conducted in the UK reported a U-shaped association between maternal Hb values and the risk of low birth weight and preterm birth[31]. A study performed in Chinese pregnant females indicated that U-shaped relationships between first-trimester Hb concentrations and the risk of preterm birth,

low birth weight, and SGA were observed without adjusting for confounding factors[32].” (Page 8, Line 25-31)

Comment 8: p. 6 Why use both CMH and binary logistic regression? Here, logistic reg. seems more appropriate. Also, if you follow my advice and leave Hb continuous, the CMH makes no sense.

Response 8: We thank you very much for your careful review. The existence of a linear connection between maternal Hb levels and neonatal outcomes was first unknown to us. Therefore, we conducted the Mantel-Haenszel Chi-square test to assess the trend between different severity of maternal anemia (non-anemia, mild anemia, moderate anemia, severe anemia groups) and rates of neonatal outcome, whose results provided evidence that there were positive relationships between the rate of preterm birth, low birth weight as well as SGA and the severity of maternal anemia. Then, according to your valuable advice on Comment 7, we left the Hb levels as a continuous variable in restricted cubic spline regression to estimate whether there are non-linear associations between Hb concentrations and risk of neonatal outcomes, and the results suggested that Hb concentrations non-linearly associated with risk of neonatal outcomes. Therefore, CMH has special significance in revealing the association between maternal anemia during early pregnancy and neonatal outcomes. We thank you again for your careful review.

Comment 9: Table 3 It would be good to show the aORs for the covariates as well as for Hb. This isn't absolutely necessary, but it would make the table more informative.

Response 9: We thank you very much for your careful review. We have reported it in Supplement Table 1.

VERSION 2 – REVIEW

REVIEWER	Dr. Peter Flom Peter Flom Consulting
REVIEW RETURNED	13-May-2023

GENERAL COMMENTS	The authors have addressed my concerns and I now recommend publication.
---

REVIEWER	Dr. Guochong Chen Soochow University Medical College, Guo-Chong Chen
REVIEW RETURNED	19-May-2023

GENERAL COMMENTS	The comments are all addressed. No further comments
---

REVIEWER	Dr. Reeta Bora Assam Medical College, Pediatrics
REVIEW RETURNED	19-May-2023

GENERAL COMMENTS	The manuscript has been refined since the previous submission and seems to be appropriate for publication now.
--

VERSION 2 – AUTHOR RESPONSE

N/A